# The Isolation, Identification and Immobilization Method of Three Novel Enzymes with Diosgenin-Producing Activity Derived from an *Aspergillus flavus*

**DOI:** 10.3390/ijms242417611

**Published:** 2023-12-18

**Authors:** Shirong Feng, Lintao Pan, Quanshun Li, Yi Zhang, Fangyuan Mou, Zhao Liu, Yuanyuan Zhang, Longfei Duan, Baofu Qin, Zhongqiu Hu

**Affiliations:** 1College of Life Sciences, Northwest A&F University, Xianyang 712100, China; fengshirong@nwafu.edu.cn (S.F.); lintaopan@nwafu.edu.cn (L.P.); liquanshun@nwafu.edu.cn (Q.L.); pre356553037@163.com (Y.Z.); fangyuanmou@nwafu.edu.cn (F.M.); liu_zhao@nwafu.edu.cn (Z.L.); zhangyuanyuan405@nwafu.edu.cn (Y.Z.); duanlongfei88888@nwafu.edu.cn (L.D.); 2College of Food Science and Engineering, Northwest A&F University, Xianyang 712100, China

**Keywords:** dioscin-glycosidase, isolation, identification, enzyme immobilization technology, stability

## Abstract

Diosgenin is an important raw material used in the synthesis of steroid drugs, and it is widely used in the pharmaceutical industry. The traditional method of producing diosgenin is through using raw materials provided via the plant *Dioscorea zingiberensis* C. H. Wright (*DZW*), which is subsequently industrially hydrolyzed using a high quantity of hydrochloric and sulfuric acids at temperatures ranging from 70 °C to 175 °C. This process results in a significant amount of unmanageable wastewater, creates issues of severe environmental pollution and consumes high quantities of energy. As an alternative, the enzymolysis of *DZW* to produce diosgenin is an environmentally and friendly method with wide-ranging prospects for its application. However, there are still only a few enzymes that are suitable for production on an industrial scale. In this study, three new key enzymes, E1, E2, and E3, with a high conversion stability of diosgenin, were isolated and identified using an enzyme-linked-substrate autography strategy. HPLC-MS/MS identification showed that E1, a 134.45 kDa protein with 1019 amino acids (AAs), is a zinc-dependent protein similar to the M16 family. E2, a 97.89 kDa protein with 910 AAs, is a type of endo-β-1,3-glucanase. E3, a 51.6 kDa protein with 476 AAs, is a type of Xaa-Pro aminopeptidase. In addition, the method to immobilize these proteins was optimized, and stability was achieved. The results show that the optimal immobilization parameters are 3.5% sodium alginate, 3.45% calcium chloride concentration, 1.4 h fixed time, and pH 8.8; and the recovery rate of enzyme activity can reach 43.98%. A level of 70.3% relative enzyme activity can be obtained after employing six cycles of the optimized technology. Compared with free enzymes, immobilized enzymes have improved stability, acid and alkaline resistance and reusability, which are conducive to large-scale industrial production.

## 1. Introduction

Diosgenin is widely used in medicine and other fields; in particular, it is used as the main precursor for a variety of steroid hormones, and is known as “medicinal gold” [1,2]. It has many pharmacological properties, such as lowering cholesterol, treating diabetes, fighting inflammation, and inhibiting cancerous tumors [3,4].

In industry, diosgenin is usually produced by the direct acid hydrolysis of *Dioscorea zingiberensis* C. H. Wright (*DZW*), in the process of which a high quantity of sulfuric acid and hydrochloric acid are used. The substances often cause significant environmental pollution. As a result, in China, traditional diosgenin processing enterprises have been forced to shut down, one after another [5]. Many cleaner and more environmentally friendly processes to produce diosgenin have been explored, including reuse of the acids, the mechanical washing of the acid hydrolysate [6], the substitution of sulfuric acid and hydrochloric acid with the other liquids [7], use of ionic liquids instead of hydrochloric acid, subcritical water extraction [8], and alcoholysis combined with a solid acid approach [9,10,11]. The disadvantage of all these methods is their high energy consumption because of the process of thermal hydrolysis, the complex preparation of ionic liquids and solid acids, or the high temperature and pressure conditions; and the secondary environmental pollution problem remains incompletely solved.

Other environmentally friendly methods of producing diosgenin have depended on the enzymes secreted by microorganisms such as *Aspergillus oryzae*, *Trichoderma reesei*, *Aspergillus niger*, *Trichoderma harzianum*, *Penicillium dioscroea*, *Lactobacillus*, *Fusarium* sp., *Curvularia lunata*, *Bacillus svelezensis*, and *Bacillus subtilis*, etc., for the cleavage of the glycosidic bonds of steroid saponins from *D. zingiberensis* on a laboratory scale [12,13,14], in which the microbial strains replace the use of commercial enzymes. Although the microbial hydrolysis methods revealed obvious merits over acidolysis without waste generation, the cultivation of microorganisms needed some essential elements such as NaNO_3_, KH_2_PO_4_, MgSO_4_, KCL, FeSO_4_, and so on [15,16], which led to difficulties with the purification of diosgenin at the next stage. Moreover, the efficiency of the microbial hydrolysis to diosgenin yield needs to be further improved [17].

A current trend is that the integrated hydrolysis methods are adopted to produce diosgenin, which consist of enzyme–acid, microorganism–acid, enzyme–microorganism, and enzyme–microorganism–acid forms [18,19,20]. No matter which approach, the target microorganisms and their secretory enzymes are very important for the industrial application to produce diosgenin. The enzymes secreted by the target microorganisms must possess the ability to hydrolyze the glycoside bond of saponins. Compared with bacteria, the compositions of fungi are more complex, the isoenzymes produced are diverse, and this may hydrolyze the types of glycosidic bonds, resulting in a relatively high production of saponins. Therefore, fungi are mostly used in the microbial transformation process [16]. However, the scope of the known and applicable fungi and isoenzymes is still limited. Endophytic fungi derived from *DZW* have always presented their hydrolysis properties of saponins, starch and cellulose to release diosgenin [21]. The endophytic fungi SY_fx2_13.2 was isolated by us from the rhizomes of *Dioscorea* peltate and contains significant properties for the production of diosgenin [22]. Because its diosgenin-producing enzymes have not yet been fully revealed and identified, their industrial application in the production of diosgenin is of great interest to us. However, these free enzymes are soluble, and their activity and stability are easily affected by the external environment, such as pH and temperature, which usually limits their reusability and increases the cost [23,24,25,26]. Thus, it is crucial to apply the immobilization of enzymes for the production of diosgenin.

The application of immobilized enzymes in the food, chemical, pharmaceutical, cosmetic and medical device industries brings important industrial and environmental advantages, such as simplifying downstream processing or continuous process operations and reducing environmental impact [27,28]. In addition, the immobilization process is both technically and economically advantageous because the stabilization can reduce the consumption of enzymes. There are many ways to immobilize enzymes, and industrial methods usually aim towards simpler and more economical methods. The most commonly used method is physical fixation [29,30]. Entrapment is a type of physical fixation method, which can fix the enzyme in the lattice structure or microcystic structure of a polymer material, while the substrate can still directly penetrate into the structure and gain contact with the enzyme. The operation process of this method is simple and convenient, and the fixing conditions are mild, which renders it suitable for its application in industrial production. Alginate has been widely used in the immobilization of enzymes due to its non-toxicity, biodegradability and biocompatibility, applied in the pharmaceutical and the food industries [31]. Therefore, it is potentially safe and effective to immobilize dioscin-glycosidase (DGA) using sodium alginate, and the produced diosgenin can then be directly used in steroid drugs. Sodium alginate is one of the cheapest alginates, easy to obtain and prepare, so it is often used as a physically fixed material [32,33,34]. In addition, it can form a gel in the presence of some bivalent cations such as Ca^2+^, and so on, which may play an important role in enzyme catalytic function [35], transforming its own shortcomings such as severe mass transfer limitation and low mechanical strength, and improving the stability of the enzyme [36,37,38].

Of course, in the physical fixation process using sodium alginate to entrap enzymes, generally the following problems remain, including low mechanical strength, low resistance to the use environment, low recovery rate of enzyme activity, and short reuse times [39,40,41]. Therefore, a clean technology to produce diosgenin using immobilized dioscin-glycosidase is needed to overcome these problems. The main objective of this study was to isolate and identify three enzymes with catalytic activity to transform saponins to diosgenin from endophytic fungus SY_fx2_13.2 [22]. In addition, their immobilization technique of sodium alginate and CaCl_2_ crosslinking was further explored and evaluated through enzyme activity, stability, acid and alkaline resistance, reusability and environmental factors (such as pH, temperature and reaction time). This work provides a new perspective of the successful industrialization application of the immobilized dioscin-glycosidases.

## 2. Results and Discussion

### 2.1. Isolation of Dioscin-Glycosidases by Fractional Precipitation with Ammonium Sulphate

#### 2.1.1. Effects of Ammonium Sulfate Concentrations on the Specific Activity of DGAs

The effects of ammonium sulfate concentrations on the specific activity of DGAs are shown in Figure 1a. It can be observed that the specific activities of DGAs rose with the increase in the ammonium sulfate concentrations (*w*/*w*) before 65%. The maximum specific activity of DGAs was achieved at 16 U/mg when the ammonium sulfate concentration was 65%, as shown in Appendix A.

The effects of the ammonium sulfate concentration on protein content are shown in Figure 1b. With the addition of ammonium sulfate, the protein contents showed a trend of gradual increase. When the concentration of ammonium sulfate was 90%, the maximum protein content was reached at 34.1 mg, but the specific activity of the enzyme declined at this point to 5.8 U/mg. This suggested that the specific activity of DGAs was inhibited by more than 65% ammonium sulfate. Therefore, 65% ammonium sulfate concentration was selected to isolate DGAs. This is similar to the results obtained when the activity of papain was at its highest when the saturation of ammonium sulfate was 60% [42].

#### 2.1.2. Effects of the Number of Salting-Out on the Specific Activity of DGAs

With the increase in the number of salting-out sessions, the specific activity of DGAs increased slightly from 16 U/mg to 17.8 U/mg, as shown in Figure 1c. However, the protein content decreased significantly (*p* < 0.0001), as shown in Appendix A. It can be seen that after three salting-out sessions, the protein content was 13.88 mg, as shown in Figure 1d. Therefore, in the experimental process, a single salting-out method was chosen to separate the enzyme.

#### 2.1.3. Effects of Gradient Ammonium Sulfate on the Specific Activity of DGAs

It was discovered that gradient ammonium sulfate plays a key role in the specific activity of DGAs. When the concentration of ammonium sulfate was 10–30%, the specific activity of DGA was 4.3–6.1 U/mg, as shown in Appendix A. When the enzyme was separated with 35% ammonium sulfate, the specific activity of the enzyme was 8.2 U/mg, which increased by 34.4%. When the ammonium sulfate concentration increased from 45% to 50%, the protein concentration increased from 19.35 mg to 24.7 mg, an increase of 27.6%. Meanwhile, the DGAs’ specific activity increased from 9.7 U/mg to 12.1 U/mg, an increase of 24.7%. It indicated that when the ammonium sulfate-graded precipitation was performed, the concentration of ammonium sulfate in the first step should be less than 35%, and the second step should be at least 45%. Otherwise, the purity and specific activity of the enzyme would be affected, and the loss of the enzyme would be larger.

The separation effects of DGAs were investigated using one step precipitation with 65% ammonium sulfate concentration and fractional precipitation with 10–30%, 30–45% and 45–65% ammonium sulfate. It can be observed in Figure 1e that the specific activity of the enzyme separated by the one-step salting-out method was 16 U/mg, and via the gradient salting-out method was 25.72 U/mg. The specific activity of DGAs obtained by gradient salting-out precipitation was much higher than that obtained by one-step precipitation. As shown in Figure 1f, the protein content obtained by one-step salting-out method was 30.46 mg, and that by the gradient salting-out was 18.89 mg. The decrease in protein content was due to the removal of impurities during the salting-out of gradient ammonium sulfate. It can be concluded that the gradient salting-out method had higher specific activity and little difference in protein content. Therefore, DGAs were isolated by the gradient salting-out method.

### 2.2. The Activity Verification and Identification of the Isolated DGAs

The verification of DGAs’ activity was investigated through a new enzyme-linked-substrate autography. The activities of the isolated DGAs were verified by a new enzyme-linked-substrate autography and are shown in Figure 2a. It can be observed that there were three protein bands which reacted with the substrate saponins, which indicated that the three proteins should possess DGAs’ activity. They were named E1, E2 and E3 in order of molecular weight from the largest to the smallest, respectively. Their molecular weights (MWs) were determined by SDS-PAGE and are shown in Figure 2b. It can be observed that the MW of E1, E2 and E3 are close to 135 kDa, 100 kDa and 48 kDa, respectively. The sequences of the three enzymes were subsequently identified by LC-MS/MS and the results are shown in Table 1.

From Table 1, it can be seen that there are 34 unique peptides identified in E1 which exist in the a-pheromone processing metallopeptidase from *Aspergillus flavus* AF70 (NCBI Reference Sequence: XP_041143655.1) with a MW of 134.45 KDa, consisting of 1187 amino acids. Its identification score is the highest in the most likely proteins in *Aspergillus f.*, with a score of 323.31. The MW of E1 is about 134.45 KDa in the result of SDS-PAGE above. In this region, there is a common structural domain named COG1025, which is the main active structural domain of the metallopeptidase, an M16 family metallopeptidase with a zinc-dependent protease, suggesting that E1 may be an isoform of insulin-degrading enzyme (Trypsinization) and cleave proteins and peptides to synthesis diosgenin. Most metalloproteases are reported to be characterized by the activation of water molecules with a divalent cation (usually zinc), the zinc in many zinc metalloproteases is replaced by some other divalent cation, and almost all cobalt- or manganese-substituted enzymes retain the catalytic activity of their zinc counterparts [43], activating the corresponding pathway function in fungi [44]. Therefore, the effect of zinc ions on the microbial conversion of diosgenin was explored in subsequent experiments.

There are 27 unique peptides identified in E2 which also exist in the glucan endo-1,3-β-D-glucosidase from *Aspergillus f.* NRRL 3357 (NCBI Reference Sequence: XP_041143323.1), with a MW of 97.89 KDa, consisting of 910 amino acids. Its identification score is the highest in the most likely proteins in *Aspergillus f.*, with a score of 310.4, which is consistent with the result of SDS-PAGE above. There is a similar structural domain PFAM17652 with many of the eukaryotic β-1,3-glucanase families between them. Endo-beta-1, 3-glucanase is a protein in the glycoside hydrolase family 81 (GH81) and an important lyase widely found in bacteria, fungi, and plants [45]. Diosgenin is widely present in the roots and stems of *DZW* mainly in the form of saponins [20], which attach glucose or rhamnose to glycosidic elements with C-O glycosidic bonds at C-3. Diosgenin is connected to the C-3 position of saponins, and then closely connected to the cell wall of some saponins-enriched plants. Therefore, E2 may possess the ability to break the cell wall structure, and directly generates saponins by breaking the C-3 bonds of position saponins bond [46]. Meanwhile, Endo-β-1, 3-glucanase has the function of hydrolyzing the β-glucoside bond in glucan, which leads to the release of diosgenin. Therefore, it suggests that E2 plays a key role in the conversion of diosgenin elements.

Analogously, there are 23 unique peptides identified in E3 which exist in the Xaa-Pro ami-nopeptidase in *Aspergillus f.* NRRL3357 (NCBI Reference Sequence: XP_041150594.1), consisting of 476 amino acids, which belongs to the aminopeptidase P family proteins. Its identification score is the highest in the most likely proteins in *Aspergillus f.*, with a score of 323.31. The MW of E3 is about 51.6 KDa in the result of SDS-PAGE above. Xaa-Pro aminopeptidase is a Mn^2+^-dependent protease commonly found in mammalian and microbial cells, usually as a membrane-bound enzyme. In the peptidase family M24 (family of formyl aminopeptidases), Xaa-pro amino peptidase is systematically expressed by *Escherichia coli*, and the divalent metal most suitable for XpmA activation/function is MnCl_2_. However, in alga TH 4, the preferred metal ion is Zn^2+^ rather than Mn^2+^ [47,48]. Therefore, the effect of zinc ions on the microbial conversion of diosgenin was explored in subsequent experiments.

Interestingly, the three enzymes have not been previously identified with DGAs’ activity, or with the effect of metal ions. Therefore, we further explored the effect of metal ions on the production of diosgenin in the solid-state fermentation of SY_fx2_13.2, shown in Figure 3. The results show that the additions of Zn^2+^, Mn^2+^ and Cu^2+^ significantly promoted the increasing activity of the enzyme in SY_fx2_13.2. It was even reported that Zn^2+^ and Mn^2+^ not only promoted microbial growth but were also required to enhance the binding affinity between the substrate and the homologous enzymes as well as to stabilize the conformation of their catalytic sites [49]. Our verification experiment shows that the appropriate addition of Zn^2+^ and Mn^2+^ is effective in increasing the yield of diosgenin. However, the yield of diosgenin decreases when over 1% metal ions are added in the medium. It is shows that excessive metal ion concentrations can inhibit the activity of the enzymes, which is consistent with previously reported results [50].

### 2.3. The Immobilization Optimization of the DGAs

#### 2.3.1. Screening and Optimization of Enzyme Protectants

The screening results of the enzyme protectants are displayed in Figure 4a. It was observed that Tween 80 was the best among all of the reported and used surfactants and protectants, including glycerol, Tween 80, NaCl, sucrose, CaCl_2_ and SDS, whether at 0, 24, 48, 72, 96 or 120 h, in which the specific activity of enzymes was significantly higher than that of the other groups (*p* < 0.01). Therefore, Tween 80 was selected as the protectant of DGAs.

The effects of Tween 80 concentrations on the specific activity of DGAs are explored and illustrated in Figure 4b, where it can be seen that the specific activity of DGAs gradually increased with the 5% gradient increase in the concentration of Tween 80 ranging from 10% to 30% (*v*/*v*) whether at 0, 24, 48, 72, 96 or 120 h. Among them, 20% Tween 80 was the best, in which the specific activity of DGAs was significantly higher than that of any of the other groups (*p* < 0.0001). In addition, 72.6% of the specific activity could be still retained whether at a concentration of 20%, 25% and 30% after 120 h. Therefore, 20% Tween 80 was selected as the protectant of DGAs in the immobilization process.

#### 2.3.2. The Immobilization Optimization Using the Box–Behnken Design

In the subsequent immobilization experiment, the Box–Behnken experiment with two levels of high and low response surface design was employed to optimized the immobilization of DGAs. Sodium alginate concentration (named A), CaCl_2_ concentration (named B), immobilization time (named C) and pH (named D) were used as independent variables, the enzyme activity recovery rate (named R1) was used as the dependent variable, and 20% Tween 80 was used as the protectant of DGAs. The design scheme and the corresponding results are listed in Table 2 on the basis of the single-factor experiment, as shown in the Appendix A.

The symbols in Table 2 and the following formula are noted as follows: R1 means immobilized enzyme active recovery rate (%); A represents sodium alginate concentration (%); B represents CaCl_2_ concentration (%); C represents immobilization time; D represents the fixed pH.
R1(%, enzyme activity recovery rate) = 39.32 + 3.04A + 1.82B + 0.4633C − 0.6833D + 2.54AB + 1.09AC + 2.14AD + 2.74BC + 2.55BD + 1.07CD − 3.95A^2^ − 3.52B^2^ − 2.39C^2^ − 1.66D^2^
(1)

The analysis results of the test results in Table 2 were shown in Table 3. R1 in Table 3 was calculated by the abovementioned Formula (1).

According to the analysis methodology of the Box–Behnken experimental design, the whole model is extremely significant when its *p*-value ≤ 0.01, and R^2^ is close to 1. It can be observed that *p*-value of the model scored below 0.0001, and R^2^ = 0.9751, indicating that the model is extremely significant. When R^2^ was adjusted to be 0.9502, 95.02% of tests could be explained by this model. It suggests that the design is successfully established. It also can be seen that factors for sodium alginate concentration (A), and CaCl_2_ concentration (B), are both extremely significant (*p* ≤ 0.0001 < 0.05). However, factors for immobilization time and pH are both not significant because their scores are 0.00178 and 0.0922, respectively, and both are over 0.05. According to the F test, the contribution rate of factors can be obtained as follows: A > B > D > C, that is, sodium alginate concentration > CaCl_2_ concentration > pH > time.

The contouring diagram and response surface diagram of the interaction of four factors A (sodium alginate concentration), B (CaCl_2_ concentration), C (immobilization time) and D (immobilization pH) are shown in Figure 5. It can be seen that within the test range of these four factors, there was a significant interaction between sodium alginate and CaCl_2_ concentrations from the contour map in Figure 5a and the 3D response surface map in 5b. This is because a network gel structure is easily achieved by sodium alginate binding with Ca^2+^. When the concentration of sodium alginate and calcium chloride are too low, the pore sizes of the gel structure are too large, and enzyme leakage will occur, leading to a low recovery rate of the enzyme activity. When their concentrations are too high, the surface of the gel will become too small, and the migration of the substrate and enzyme will be prevented, which leads to the reduction in the recovery rate of enzyme activity. Therefore, there is an optimal concentration for sodium alginate or CaCl_2_. Analogously, there are the descending levels of interactions between CaCl_2_ and immobilization time, between pH and CaCl_2_, between sodium alginate concentration and pH, between calcium chloride concentration and immobilization time, between immobilization time and pH, and between fixed time and pH, as shown in the figures from Figure 5c–l. The optimal concentration for sodium alginate or calcium chloride, the fixed time and pH were predicted using Design-Expert 13 software, which are sodium alginate concentration of 3.5%, CaCl_2_ concentration of 3.45%, immobilized time of 1.4 h, pH of 8. 8, and an enzyme activity recovery rate of 43.98%, under these conditions. The verification experiments were conducted three times under the optimal conditions and the results showed that the recovery rates of the immobilized enzymes were 44.12%, 43.35% and 44.47%, which is in good agreement with the predicted 43.98%. The optimized process parameters can effectively improve the enzyme activity recovery rate. Combined with the results of this study, the formation of sodium alginate gel is achieved by binding with Ca^2+^ to form a network structure, and the enzyme is wrapped in the structure.

#### 2.3.3. Stability Test of Immobilized DGAs

The immobilization method altered the enzyme’s pH and temperature stability, thereby increasing its stability to environmental pH and temperature changes. As illustrated in Figure 6a, the relative activity of the immobilized DGAs could be maintained at over 89.57% when the pH ranged from 6.5 to 8.5, which was higher than that of the free DGAs. In particular, the optimal pH of the immobilized DGAs was at 7.5, not the optimal 7.0 of the free DGAs. This may be due to the protective effects of sodium alginate gel and the provided protective agent, and negative effects of irrelevant ions on enzyme activity. These results are supported by a similar study [51].

The effects of the immobilization method on the temperature stability of DGAs are depicted in Figure 6b. It can be observed that the relative activity of the immobilized DGAs could be maintained at over 77.36% when the temperature ranged from 30 °C to 80 °C, which was higher than that of the free DGAs. In addition, the optimal temperature of the former drifted to 60 °C from the 50 °C optimal temperature of the latter. This should be attributed to the protective effect of the network structure formed by sodium alginate and CaCl_2_; in addition, the impact of high temperatures on the enzyme’s active center was notably reduced by the network structure [52].

The effect of the immobilization method on the reaction time stability of DGAs is shown in Figure 6c. It can be seen that the optimum reaction time of immobilized DGAs was 14 h, increasing the optimum reaction time of the free DGAs by 2 h. Moreover, the relative activity of the immobilized DGAs declined more slowly than that of the free DGAs after 14 h. This may have been caused by the embedded network structure which formed with sodium alginate and calcium chloride, alleviating the inhibition of the produced diosgenin.

The most prominent advantage of immobilized enzymes over their free counterparts is their capacity for reusability [53]. The figures for recycling of immobilized DGAs were determined and displayed in Figure 6d. With the increase in time, the relative enzyme activity of the immobilized DGAs gradually decreased from 100% to 70.30% over six consecutive uses. It means that, fundamentally, the immobilized DGAs can still basically meet the needs of production over six cycles. This activity decline in the immobilized DGAs may be attributed to the relatively weak binding forces, primarily non-covalent bonds, which may lead to the shedding and loss of DGAs from the immobilization carrier with the progress of the enzyme reaction. Furthermore, the reduction in activity can be ascribed to the hydrophilic nature of sodium alginate. Over time, the pores within the immobilization matrix tend to expand, resulting in increased enzyme leakage and subsequently a decrease in enzyme activity, which is consistent with similar, previous reports [48].

Effective reusability has the potential to reduce the demand for free enzymes in industrial production, consequently leading to reduced production costs. It should be noted that the activity of the immobilized enzyme gradually diminishes with each successive reuse cycle; a certain amount of DGAs’ supplementation should be essential according to the law of enzymatic activity decline.

## 3. Materials and Methods

### 3.1. Materials and Chemicals

The strain SY_fX2_13.2 (Conservation No. 11517) used in this experiment belongs to *Aspergillus flavus,* a kind of endophyticus of *Dioscorea* peltata, which was isolated from the rhizomes of *Dioscorea* peltata and kept in our laboratory. The rhizomes of DZW were procured from the National Phytochemical Engineering Research Center (Western Division).

The standard products of diosgenin and saponin (purity ≥98%) were purchased from the China National Institute for the Control of Pharmaceutical and Biological Products (NICPBP, Beijing, China). Pure water was produced by UPHW-I-90Z ultrapure water equipment (Shanghai Hitech Instruments, Shanghai, China). Analytical grade alcohol was purchased from the Nanjing Chemical Reagent Co. Ltd. (Nanjing, China). Analytical grade sodium chloride, sodium hydroxide, CaCl_2_, sucrose, potassium dihydrogen phosphate, and anhydrous magnesium sulfate were obtained from Tianjin Bodi Chemical Co. Ltd. (Tianjin, China). The beef paste, yeast powder, and potato dextrose agar medium (PDA) were obtained from Beijing Aobox Biotechnology Co. Ltd. (Beijing, China). Analytically pure petroleum ether, ethyl acetate, and chloroform reagents were procured from Sigma-Aldrich (Shanghai, China).

### 3.2. Primary Culture and Liquid Fermentation of the Strain

The endophytic *Aspergillus flavus* SY_fx2_13.2 strain was activated on potato dextrose agar medium (PDA) for 3–6 days, following which the spores were rinsed off and loaded into conical flasks. Subsequently, the spores were dispersed by the added glass beads and shaken well to form a single-spore suspension. The resulting single-spore suspension was inoculated into the primary seed liquid medium (0.45% sucrose, 0.1% beef paste, 0.025% KCL, 0.075% K_2_HPO_4_·3H_2_O, 0.025% MgSO_4_·7H_2_O, 0.71% *Dioscorea rhizome* powder) at a rate of 5% and then cultivated for one day under aerobic conditions at 33 °C and 120 rpm. The obtained culture was recultured in the same medium above with 5% (*v*/*v*) at 33 °C and 160 rpm in a shaker for 48 h.

The resultant culture was inoculated into the designed fermentation medium which consisted of 1.79% sucrose, 2.08% beef paste, 0.05% KCL, 0.15% K_2_HPO_4_·3H_2_O, 0.05% MgSO_4_·7H_2_O, 0.001% FeSO_4_·12H_2_O, and 1.41% *Dioscorea rhizome* powder, at a rate of 5%, and then cultivated for 5 days under aerobic conditions at 33 °C and 110 rpm.

### 3.3. Preparation of Crude Enzyme Solution

The cells were harvested by filtration through nylon cloth and ground with liquid nitrogen to be broken and formed into a powder. The bacteriophage powder was blended with the previously prepared acetic acid–sodium acetate buffer solution (0.02 M, pH 5.0) at a ratio of 1:5 (*w*/*v*). The mixture was agitated for a duration of 15 min, followed by centrifugation at 10,000 rpm for 20 min at a temperature of 4 °C. The supernatant was collected to extract the crude enzyme which can hydrolyze saponins into diosgenin.

### 3.4. Isolation and Optimization of DGAs

The above crude enzyme solution was subjected to three stages of isolation by the ammonium sulfate precipitation method to obtain the optimal separation conditions [54].

In the first stage, a certain amount of solid ammonium sulfate was separately added in 14 portions to the crude enzyme solution, 50 mL each, and maintained at 10%, 20%, 30%, 40%, 45%, 50%, 55%, 60%, 65%, 70%, 75%, 80%, 85%, and 90% (*w*/*v*) saturation, which were mixed well and left to stand for 12 h at 4 °C. Subsequently, the resultant mixtures were centrifuged at 5180× *g* for 20 min. The resultant precipitates were individually reconstituted in 0.02 M acetate–sodium acetate buffer at pH 5.0. Subsequently, they were subjected to dialysis within dialysis bags featuring a molecular weight cut-off (MWCO) range of 8000–14,000 Da. This dialysis process was carried out with the assistance of a magnetic stirrer, spanning a duration of 12 h, and the procedure was repeated twice. The retention fluids were fixed with 0.02 M pH 5.0 acetic acid–sodium acetate buffer to 10 mL, respectively.

In the second stage, three portions of crude enzyme solution were taken, numbered 1–3. A certain amount of solid ammonium sulfate was separately added in 3 portions of crude enzyme solution and maintained at 65% (*w*/*v*) saturation. These were centrifuged at 10,000 rpm for 20 min and discarded; the supernatant was discarded at 4 °C. The samples of No. 2 and No. 3 were re-dissolved with 50 mL buffer for a second salting-out, and then were centrifuged and the supernatant was discarded. No. 3 sample was salted out for the third time, and the subsequent steps were the same as above. No. 1, 2, and 3 samples were re-dissolved with 5 mL buffer, and the triplicate enzyme solution was dialyzed to a constant volume of 10 mL.

In the third stage, three parts of the crude enzyme solution were taken. Firstly, a certain amount of solid ammonium sulfate was added to enzyme solution No. 1, which was maintained at 30% saturation and left to stand for 4 h, then centrifuged at 10,000 rpm for 20 min. The precipitate was discarded, and solid ammonium sulphate was continuously added to the supernatant to saturate it to 45% and then left to stand for 4 h. After, it was centrifuged at 10,000 rpm for 20 min. The precipitate was discarded, which maintained 65% saturation. It was left to stand for 8 h at 4 °C, then centrifuged and the supernatant was discarded. The scheme of this work is shown in Figure 7. In enzyme solution No. 2, ammonium sulfate was added to the enzyme liquor, which maintained a final 65% concentration. This was mixed well and left to stand for 12 h at 4 °C. The mixtures were centrifuged at 10,000 rpm for 20 min, and the supernatant was discarded. The precipitates obtained from centrifugation of No. 1 and No. 2 were redissolved with acetic acid–sodium acetate buffer, dialyzed and concentrated to 10 mL.

Comparing the above methods, the ammonium sulfate precipitation method with relatively high specific activity was selected for the enzyme isolation.

### 3.5. Identification of DGAs

After ammonium sulfate fractional precipitation, the enzyme liquid was filtered through an aqueous 0.45 μm membrane, and the resulting enzyme solution was concentrated 5-fold by ultrafiltration (10 kDa, 4000 rpm, 4 °C).

The activity of the isolated proteins was preliminarily analyzed by Native-PAGE (N-PAGE) using 12% (*w*/*v*) native polyacrylamide gels according to the method described by Schagger and Von [55]. The N-PAGE was cut from the center, half immersed in saponin solution (20 mg/mL) and the other half was stained with Coomassie Brilliant Blue R-250, then placed on an orbital oscillator at 4 °C. The diosgenin precipitated on the gel to form bands. Subsequently, the active target enzyme hydrolyzed, the two gels were compared, the corresponding target bands were cut, and identity confirmation of the target enzyme took place using HPLC/MS-MS analysis.

The gel strip with the target-containing protein was cut from N-PAGE and soaked in 0.02 M, pH 5.0 acetic acid–sodium acetate buffer solution at 4 °C, according to the ratio of each piece of recycled gum strips plus 5 mL of buffer. Electrophoresis was then repeated to obtain the DGAs’ activity of the gum strips, and the strips were then all collected (30 pieces of gum). All strips obtained were slowly ground under ice bath conditions and then transferred to a conical flask (250 mL). The conical flask was placed at 4 °C for 24 h, during which time it was shaken several times to ensure the solubilization of the DGAs. The recovered solution with gel was filtered through nylon cloth and then through a 0.45 μm microporous filter membrane. The enzyme solution was concentrated 20-fold by ultrafiltration (10 kDa) and centrifugation (4000 rpm) at 4 °C. The molecular weight distribution of the isolated and purified proteins was initially assessed through SDS-PAGE analysis of the enzyme solution recovered in the secondary stage, using a 12% (*w*/*v*) polyacrylamide gel. The molecular weight of each specific target protein was determined via linear regression, correlating the logarithm of the molecular weight to the relative mobility of standard proteins in the markers employed [56].

### 3.6. HPLC/MS-MS Identification of DGAs

Some modifications were made to the previously reported method for LC-MS/MS [57]. The gel strip containing the protein of interest was excised from the Native-PAGE, and subsequently subjected to decolorization using 50% ethanol until complete decolorization was achieved. It was then dried using a vacuum centrifuge and subjected to an overnight digestion in a trypsin solution with a concentration of 10 ng/μL, which was prepared using a 100 mM sodium bicarbonate aqueous solution. The resulting peptides were extracted once using a solution containing 50% acetonitrile and 5% formic acid (FA), and then twice with 100% acetonitrile. The isolated solutions were amalgamated and thoroughly desiccated using a vacuum centrifuge. The identification of the extracted peptides was performed using an Easy nLC coupled with a Q Exactive HF-X mass spectrometer (Thermo Fisher Scientific, Waltham, MA, USA). The peptide mixture was loaded onto a Phenomenex Aqua RP-C18 column (50 μm × 15 cm, 2 μm). Mobile phase A was 0.1% formic acid aqueous solution (*w*/*w*). Mobile phase B was 0.1% formic acid, and 80% acetonitrile aqueous solution (*w*/*w*), and the interflow technique controlled the flow rate at 500 nL/min. The elution gradient was as follows: Phase 1: 0–36 min, 2% B-35% B; Phase 2: 36–45 min, 35% B-45% B; Phase 3: 45–47 min, 35% B-45% B; Phase 4: 0–35 min, 2% B-35% B; Phase 5: 0–35 min, 35% B-45% B; Stage 3: 45–47 min, 45% B-80% B; Stage 4: 47–50 min, 80% B; Stage 5: 50–51 min, 80% B-2% B; Stage 6: 51–60 min, 2% B. The MS/MS analysis was set to positive ionization mode for 60 min. The precursor ion scan range was 400–1800 *m*/*z*. The MS resolution was set to 60,000 *m*/*z* 200 and 15,000 *m*/*z* 200 for MS1 and MS2, respectively. The maximum self-gain control (AGC) settings were configured at 1 × 10^6^ for MS1 and 1 e^5^ for MS2, respectively. The maximal injection time was set at 30 ms for MS1 and 50 ms for MS2, respectively. The dissociation method employed was higher-energy collision dissociation (HCD), with a normalized collision energy of 30. The isolation window was 1.6 Th. MS data were searched against the UniProtKB *Aspergillus_flavus* database (65536 entries, downloaded 5 June 2023) using MaxQuant software version 2.0.3.1. We established an initial maximum permissible mass tolerance of 4.5 parts per million (ppm) for precursor ions and 20 ppm for fragment ions in the context of HCD MS/MS spectra. The quest for peptide identification adhered to the trypsin cleavage rule and permitted a maximum of two absent cleavage sites. The threshold for the global false discovery rate regarding both peptide and protein identification was rigorously set at 0.01.

### 3.7. Analysis of DGAs Activity

The quantification of diosgenin content was conducted using ultraviolet-visible (UV-3600 Shimadzu, Kyoto, Japan) spectrophotometry, with absorbance measurements performed at a wavelength of 460 nm. The DGAs activities unit is represented by U, and under specific conditions (optimal temperature and pH), the amount of enzyme required to catalyze the production of 1 μmol of diosgenin from saponins substrates per minute, is 1U = 1 μmol/min.

The protein content of enzyme solution was obtained using the Bradford method [55]. In addition, 5 mL of each enzyme solution was taken and reacted with an equal volume of 20 mg/mL of saponins substrate for 12 h. After the reaction was terminated by a boiling water bath, 10 mL of petroleum ether was added to the extract for 3 h. The operation was repeated three times, and all extracts were transferred to Petri dishes.

After the petroleum ether was completely dried, the product was dissolved in trichloromethane to 10 mL and then analyzed by TLC. The unfolding agent was cyclohexane/ethyl acetate (4:1, *v*/*v*), and the color rendering agent was 10% ethanol sulfate solution. The enzymatic products were solubilized by Lieberman–Burchard solution and the volume was fixed with trichloromethane to 10 mL, which was terminated in a 50 °C boiling water bath for 60 min. The reaction solutions were measured at 460 nm for absorbance value. The reaction solutions were measured at 460 nm for absorbance value and the concentration of the diosgenin was calculated by the standard curve of diosgenin to detect the activity of DGAs.

The specific activity of DGAs activity was determined and calculated according to the following Formula (2):(2)$$specific\;activity\;(U/mg)=\frac{enzyme\;activity}{w}$$

In the formula, W: protein content (*mg*).

### 3.8. Screening and Optimization of Protective Agents for Enzyme Immobilization

The prepared DGAs were divided into seven portions of 40 mL each in tubes, and equal proportions of glycerol, Tween 80, NaCl, sucrose, CaCl_2_, and SDS were separately added in the first six tubes; their final concentrations were 20% and the seventh tube was a blank control. They were mixed well and continuously reacted with an equal volume of 20 mg/mL of saponins substrate solution. The specific activities of DGAs were determined at 0 h, 24 h, 48 h, 72 h, 96 h and 120 h, respectively. The specific activities of each group were determined to screen for the optimal enzyme protectants. The best protectant selected from the previous step was further optimized by adding individual proportions of 10%, 15%, 20%, 25% and 30% of it in 40 mL of each enzyme solution. The specific activities of DGAs were determined at the different times as above.

### 3.9. Preparation Conditions and Optimization of Immobilized DGAs

The given DGAs were uniformly blended with 2%–4% aqueous sodium alginate solutions in a volumetric ratio of 1:3 (*v*/*v*). The resulting mixture was drawn up into a 5 mL injector and then gradually introduced into a CaCl_2_ solution from a height of 5 cm above the liquid level. Sodium alginate was solidified at 4 °C in a refrigerator overnight. Then, they were washed with deionized water and finally stored at 4 °C after the gel microspheres were observed [52,58].

To establish the low and high levels of individual variables, a single-factor test was conducted and is shown in the Appendix A. Subsequently, the Box–Behnken design was employed by Design Expert 13 software to investigate the impact of diverse factors on the recovery rate of DGA activity [59], in which sodium alginate concentration (A), CaCl_2_ concentration (B), immobilization time (C) and pH (D) were regarded as the independent variables, the enzyme activity recovery rate (R1) was used as the dependent variable. The recovery rate of DGAs activity (R1) was determined by the method as above-mentioned through calculating the average value of three independent measurements. The design of all factors and levels selected according to the results of single-factor tests is shown in Table 4.

### 3.10. Stability Test of Immobilized DGAs

#### 3.10.1. Effect of pH Values on the Activities of the Immobilized DGAs

A certain amount of immobilized DGAs and free enzyme were added to saponins for reaction, and the pH was adjusted to 6.5, 7.0, 7.5, 8.0, 8.5 with aceto–sodium acetate buffer solution, respectively. The free DGAs were used as controls. The reaction was reacted with an equal volume of 20 mg/mL of saponins substrate solution and carried out at 60 °C for 14 h in a shaking table. The reactions were terminated in a boiling water bath. DGAs activity was determined and expressed as relative enzyme activity (%) to that of free DGAs.

#### 3.10.2. Effect of Temperatures on the Activities of the Immobilized DGAs

The effect of temperatures on the activities of the immobilized DGAs was assessed at various temperatures ranging from 30 °C to 80 °C according to the previously described method. pH was adjusted to 7.5 and reacted with an equal volume of 20 mg/mL of saponins substrate solution for 14 h. The free DGAs were used as the controls. The relative enzyme activity of the immobilized DGAs (%) to that of the free DGAs at the different temperatures was calculated.

#### 3.10.3. Effect of the Enzymatic Time on the Activities of the Immobilized DGAs

The reaction solutions as above were prepared and carried out under the optimal conditions of pH 7.5 and 60 °C, which were terminated at 9 h, 10 h, 11 h, 12 h, 13 h, 14 h, 15 h and 16 h in a boiling water bath, respectively. The free DGAs were used as the controls. The relative enzyme activity of the immobilized DGAs (%) to that of free DGAs at the different time points was calculated.

#### 3.10.4. Recycling of Immobilized DGAs

The prepared immobilized DGAs were continuously reacted with a given amount of saponins at pH 7.5 and 60 °C with a stirring speed of 120 rpm/min for 6 cycles, for 14 h each cycle. The activities of DGAs in the first reaction cycle were regarded as 100%, and the relative activities of the subsequent reaction cycles were calculated relative to that of the first reaction cycle.

### 3.11. Recovery of Immobilized DGAs Activity

Recovery of immobilized DGAs activity was determined and calculated according to the following Formula (3):(3)$$Recovery\;of\;immobilized\;enzyme\;activity\;\left(\%\right)=\frac{total\;immobilized\;enzyme\;activity}{total\;free\;enzyme\;activity}\times 100\%$$

## 4. Conclusions

In this work, three DGAs with an active role in the production of diosgenin were isolated and identified from the fermentation products of the endophytic fungus strain SY_fx2_13.2 by means of tertiary precipitation, N-PAGE coupled with enzyme-linked-substrate autography, which has never before been characterized as being able to convert saponins. In order to achieve industrial applications, their immobilization method was explored by the Box–Behnken design and the optimal technology parameters were obtained, namely, a sodium alginate concentration of 3.5%, CaCl_2_ concentration of 3.45%, immobilized time of 1.4 h, and pH of 8. 8, which allowed the recovery rate of enzyme activity to reach 43.98%. After six cycles, a 70.3% relative enzyme activity of the immobilized DGAs could be obtained using the proposed method. Compared with free enzymes, immobilized DGAs have better stability, acid and alkaline resistance and reusability, which are all conducive to large-scale industrial production. The proposed immobilization method provides a good strategy for preparing immobilization biocatalysts, which is expected to bring potential economic and ecological benefits.

This work also provides an environmentally friendly, green and energy efficient method to produce diosgenin by mild enzymatic hydrolysis using the three DGAs. In addition, the enzyme-linked-substrate autography strategy provides a reliable and effective enzyme identification model for the discovery of better DGAs or other specific functional enzymes.

## Figures and Tables

**Figure 1 ijms-24-17611-f001:**
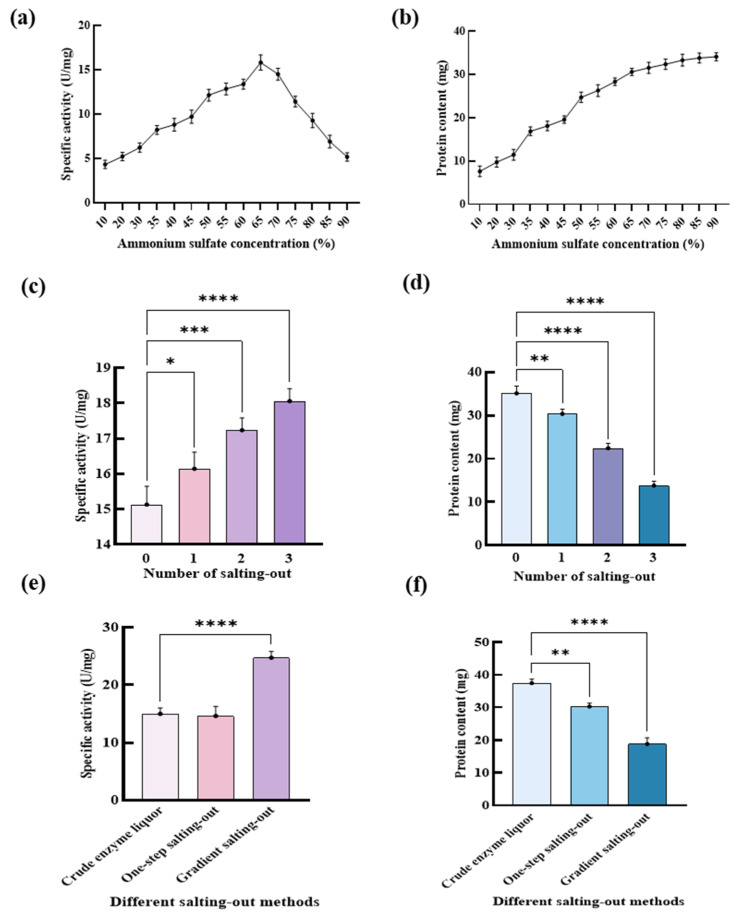
(**a**) Influence of ammonium sulfate concentration on specific enzyme activity. (**b**) The effect of ammonium sulfate concentration on protein content. (**c**) Influence of salting-out times on specific enzyme activity. (**d**) Influence of salting-out times on protein content. (**e**) Effects of different salting-out methods on specific enzyme activity. (**f**) Effects of different salting-out methods on protein content. (* *p* < 0.05; ** *p* < 0.01; *** *p* < 0.001; **** *p* < 0.0001.)

**Figure 2 ijms-24-17611-f002:**
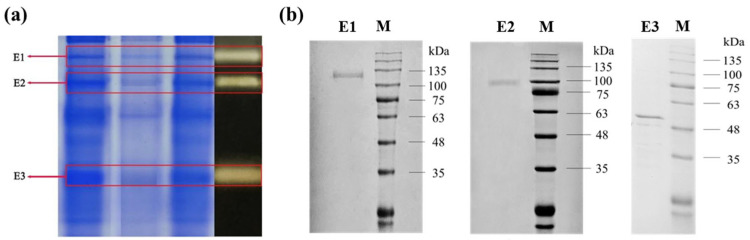
The activity verification and molecular weight analysis of the isolated DGAs. (**a**) Verified dioscin-glycosidases by enzyme-linked-substrate autography. (**b**) The SDS-PAGE results of purified E1, E2 and E3. Lane 1, 2 and 3: the enzyme solution of E1, E2 and E3 after Native-PAGE, respectively; M: protein marker.

**Figure 3 ijms-24-17611-f003:**
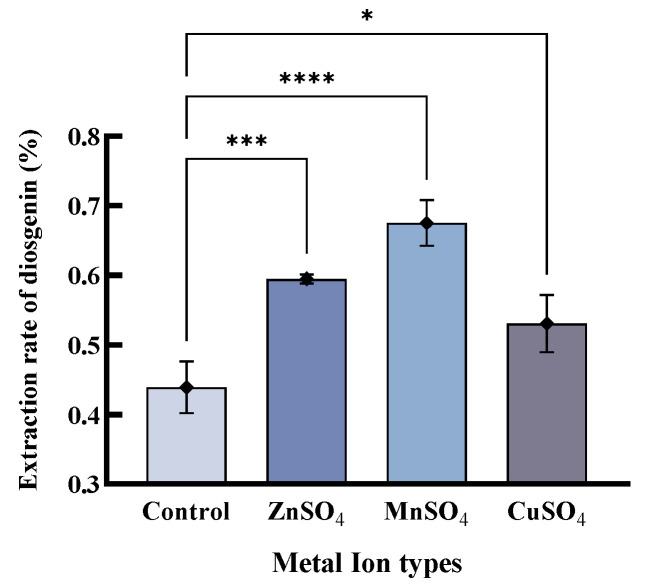
Effect of metal ions on the production of diosgenin by solid-state fermentation. (Control: base medium.) (* *p* < 0.05; *** *p* < 0.001; **** *p* < 0.0001.)

**Figure 4 ijms-24-17611-f004:**
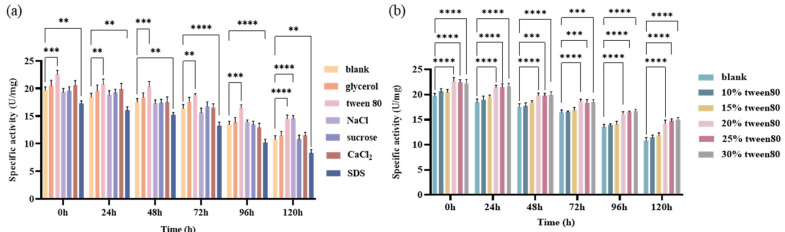
Screening and optimization of enzyme protectants. (**a**) Effect of the addition of different protective agents on the specific activity of DGAs. (**b**) Effect of Tween 80 concentration on the specific activity of DGAs. (*** p* < 0.01; *** *p* < 0.001; **** *p* < 0.0001.)

**Figure 5 ijms-24-17611-f005:**
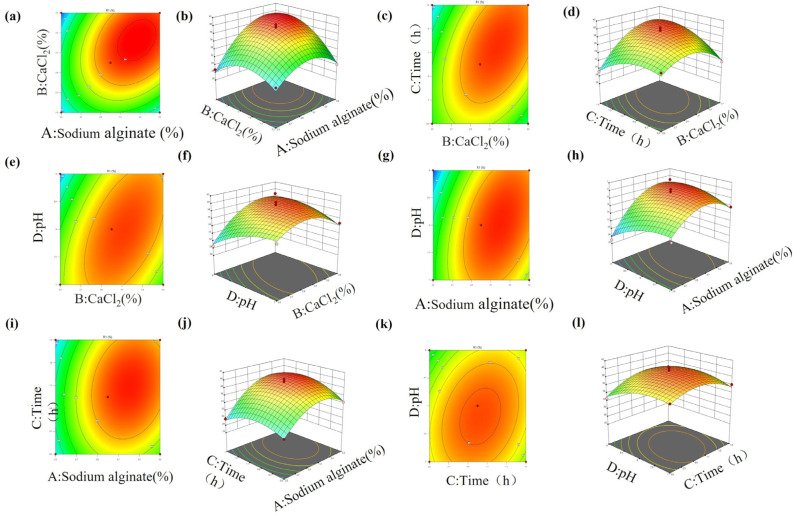
Response surface (3D) and contour plots showing the effect and interaction of the five main factors on the recovery immobilized enzyme. (**a**,**b**) Interaction between sodium alginate concentration and CaCl_2_ concentrations. (**c**,**d**) Interaction between time and CaCl_2_ concentrations. (**e**,**f**) Interaction between CaCl_2_ concentrations and pH. (**g**,**h**) Interaction between sodium alginate and pH. (**i**,**j**) Interaction between sodium alginate and time. (**k**,**l**) Interaction between time and pH.

**Figure 6 ijms-24-17611-f006:**
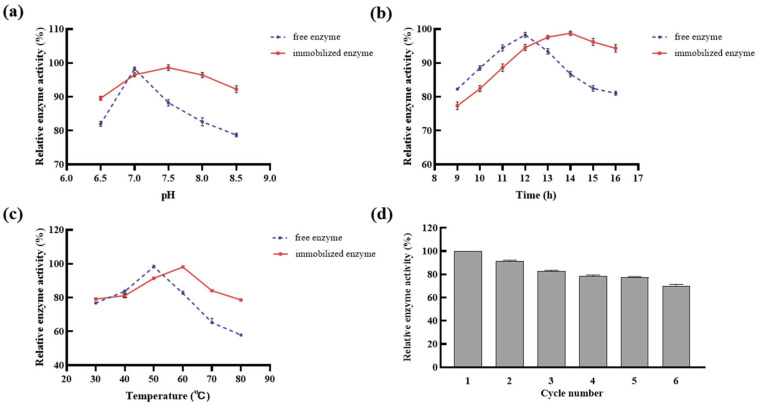
Stability of immobilized enzymes. (**a**) pH stability. (**b**) Temperature stability. (**c**) Time stability. (**d**) Stability of the number of cycles.

**Figure 7 ijms-24-17611-f007:**
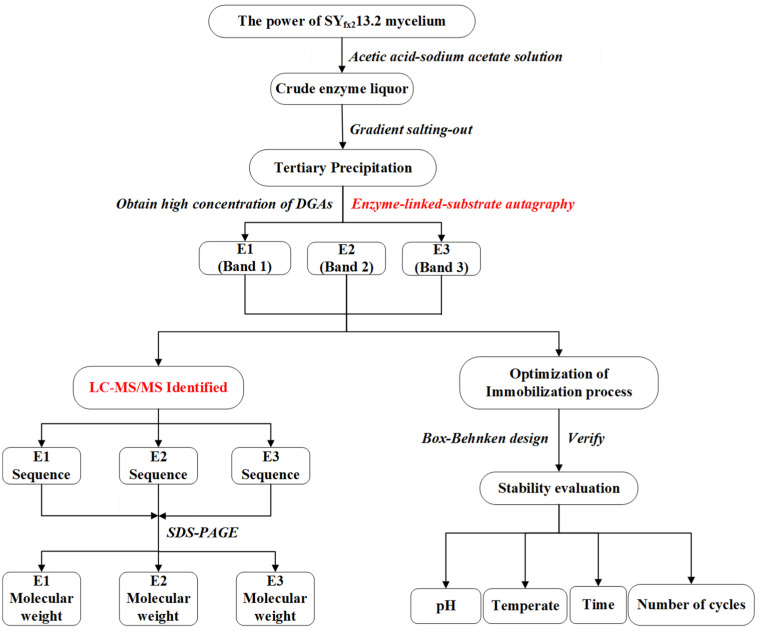
The scheme of this work.

**Table 1 ijms-24-17611-t001:** Identification of DGAs.

Variable	E1	E2	E3
Similar protein	A-pheromone processing metallopeptidase Ste23	glucan endo-1,3-beta-D-glucosidase	Xaa-Pro aminopeptidase
Regulation gene ID	64846173	64845841	64853314
score	323.31	310.4	323.31
Unique peptides	34	27	23
Sequence coverage (%)	35.8	38.2	61.5

**Table 2 ijms-24-17611-t002:** BBD experimental table and the corresponding result.

Number	Factor				
	A	B	C	D	Enzyme Activity Recovery Rate (%)
1	0	0	0	0	40.31
2	−1	0	0	−1	32.34
3	0	0	0	0	39.08
4	0	1	−1	0	31.68
5	1	1	0	0	38.7
6	1	0	0	1	38.89
7	1	0	−1	0	34.26
8	1	−1	0	0	30.2
9	−1	−1	0	0	30.4
10	0	1	0	1	38.65
11	0	1	1	0	37.21
12	−1	0	1	0	29.7
13	−1	0	−1	0	30.65
14	0	1	0	−1	34.73
15	0	−1	0	1	28.6
16	1	0	1	0	37.66
17	−1	1	0	0	28.76
18	0	−1	−1	0	34.61
19	−1	0	0	1	27.01
20	0	0	0	0	39.05
21	0	0	−1	1	32.67
22	0	0	0	0	38.5
23	0	0	1	1	36.31
24	0	−1	1	0	29.2
25	0	0	0	0	39.67
26	1	0	0	−1	35.67
27	0	0	−1	−1	36.71
28	0	0	1	−1	36.06
29	0	−1	0	−1	34.88

**Table 3 ijms-24-17611-t003:** Variance analysis.

Source	Sum of Squares	Degree of Freedom	Mean Square	F-Value	*p*-Value
Model	431.80	14	30.84	39.12	<0.0001
A	111.14	1	111.14	140.98	<0.0001
B	39.75	1	39.75	50.42	<0.0001
C	2.58	1	2.58	3.27	0.0922
D	5.69	1	5.69	7.21	0.0178
AB	25.70	1	25.70	32.61	<0.0001
AC	4.73	1	4.73	6.00	0.0281
AD	18.28	1	18.28	23.18	0.0003
BC	29.92	1	29.92	37.95	<0.0001
BD	26.01	1	26.01	32.99	<0.0001
CD	4.60	1	4.60	5.84	0.0299
A^2^	100.98	1	100.98	128.09	<0.0001
B^2^	80.51	1	80.51	102.13	<0.0001
C^2^	36.91	1	36.91	46.83	<0.0001
D^2^	17.89	1	17.89	22.69	0.0003
Residual	11.04	14	0.7883		
Lack of Fit	9.13	10	0.9131	1.92	0.2777
Error	1.91	4	0.4764		
Total	442.84	28			

**Table 4 ijms-24-17611-t004:** Factors and code levels of variables for the Box–Behnken design.

Variable	Unit	Code Level-Value	Factor Value
Sodium alginate concentration	%	−1, 0, 1	2.5; 3; 3.5
CaCl_2_ concentration	%	−1, 0, 1	2.5; 3; 3.5
Immobilization time	h	−1, 0, 1	0.5; 1; 1.5
pH	-	−1, 0, 1	7.0; 8; 9.0

## Data Availability

All experimental data are included in this paper and the Appendix A.

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
