# Peer review of "The Isolation, Identification and Immobilization Method of Three Novel Enzymes with Diosgenin-Producing Activity Derived from an Aspergillus flavus"

_ijms, 2023, doi:10.3390/ijms242417611_

Round 1

Reviewer 1 Report

Comments and Suggestions for Authors

The manuscript of Feng et al. addresses the isolation of three enzymes able to hydrolyze dioscin, a steroid saponin, to obtain diosgenin. These enzymes with dioscin glycosidase activity were identified by HPLC-MS/MS and immobilized by entrapment in a calcium alginate matrix, the immobilization conditions being optimized by experimental design (Box-Behnken method). The authors claim an overall 40% recovered enzyme activity following immobilization and 70% remanent activity after 6 reuse cycles. Although the topic is of scientific interest and the results are valuable, I cannot recommend the publication of the manuscript as it needs, in my opinion, a thorough revision. Particularly, the discussion of the results was not carried out in an obvious and scientifically appropriate manner. I will point out several issues that should be resolved, but the whole manuscript needs major improvement.

1. Please delete “technology” from the title. This work reports the results of immobilization experiments, it’s a long way up to a technology.

2. The introduction part is not appropriate, essential data are missing. The utilization of dioscin glycosidases for the hydrolysis of DZW saponins is mentioned in only one phrase (lines 48-50), indicating 4 references. Please extend this part by adding other references (there are many), like as Xiang et al., 2018 (https://doi.org/10.1007/s00253-018-9030-5), Chen et al., 2018 (https://doi.org/10.1016/j.steroids.2018.05.005), and particularly the recent review of Zhang et al. (https://doi.org/10.1016/j.steroids.2023.109181). Moreover, the results reported by other research groups must be briefly presented, allowing a comparison with their own results (which must be performed in the discussion section).

3. In the introduction part, all references  (#18 to #21) supporting the limitations of soluble (native) enzymes utilization and their overcoming by immobilization were selected from a single enzyme class, the lipases. Instead, the authors must cite review articles or book chapters that discuss the advantages of immobilizing enzymes (mainly for industrial applications) in a more general manner. The immobilization by entrapment, particularly in alginate matrices, must be also described based on literature data, highlighting its advantages compared to other immobilization methods for the immobilization of the studied enzymes. I recommend completely rewriting and enlarging this part.

4. Results and discussion, page 2. Please add a reaction scheme (or a more comprehensive scheme, also including the isolation and immobilization steps) to provide an overview of the accomplished work. Also, please add at the beginning of every subchapter a phrase summarizing and justifying the methodology selected for the experimental work described in that section.

5. Page 2, Subchapter “Isolation optimization of dioscin glycosidases” (this title must be changed, maybe to “Isolation of dioscin glycosidases by fractional precipitation with ammonium sulphate”, maybe it would also be useful to number the subchapters as 2.1, 2.2, etc.) Please add a table presenting the activity, specific activity and total activity recovery yield for every isolation and purification step. Otherwise, it is difficult for the reader to evaluate the real efficiency of this approach.

6. I cannot identify the term “gel imaging autography”. As I understand, N-PAGE was used as a preliminary check of dioscin glycosidase activity in the fraction spots, but please explain it in the text.

7. The authors must disclose what exactly means “protein concentration” determined in the presence of so-called “protective agents” and the significance of its determination for the immobilization of the enzyme (being included in the immobilization optimization chapter).

8. Materials and methods part and throughout the manuscript. Please calculate the enzymatic activity units as micromoles/min, as commonly used, not as micrograms/min. The definition given at page 14, lines 529-531 “1 U is the amount of 1 microgram of diosgenin produced per minute per milliliter of dioscin-glycosidase solution to be tested by reacting with the saponin substrate” is not correct. As it was defined long time ago, the standard unit of enzyme activity is the enzyme amount which will catalyze the transformation of 1 micromole of the substrate per minute under standard conditions (Eur. J. Biochem. 1979, 97, 319-320). So, the activity of the enzyme in liquid form (solution) must be expressed as U/ml, while the activity of a solid enzyme preparate as U/mg.

9. The manuscript contains many syntax, grammar, and spelling errors. Please revise it comprehensively and also be consistent and clear in the discussion part. The following examples represent only a selection, it is impossible for me to discuss all.

“…DZW was commonly used DZW…” (abstract)

“…severe environment-polluted problem…” (abstract)

“…have tried to develop a cleaning process for the production of diosgenin…” (lines 44-45)

“… the low resistance to the use environment…” (line 56)

“Most metalloproteases are reported to be characterized by the activation of water molecules  with a divalent cation (usually zinc), the zinc in many zinc metalloproteases is replaced by some other divalent cation, and almost all cobalt- or manganese-substituted enzymes  retain the catalytic activity of their zinc counterparts[27], activates the corresponding pathway function in fungi” (lines 133-138).

“Diosgenin is widely found in DZW mainly in the form of steroidal  saponins in their rhizomes[29], which attach glucose or rhamnose to glycosidic elements  with C-O glycosidic bonds at C-3. Diosgenin is widely found in DZW mainly in the form of steroidal saponins in their rhizomes, which attach glucose or rhamnose to glycosidic  elements with C-O glycosidic bonds” (lines 145-149, the same text repeated)

“The reaction can  be any N-terminal amino acid linked to proline. Xaa-pro amino peptidase systematically  expressed by Escherichia coli, which the divalent metal most suitable for XpmA activation/function was MnCl2” (lines 162-165)

Figure 3. ZnSO4, etc., are not metal ion types, they are salts. Maybe Zn2+, etc.

Line 224. “R1: Extraction rate of diosgenin”. In Table 2, the R1 symbol cannot be found. R1 is depicted in equation (1) as enzyme activity recovery rate, but in the explanation of the symbols (line 256) Y is used as symbol for the same variable.

“…immobilized time…” (line 274)

“…the fixation of diosgenin…” (line 283)

“…while the immobilized amylase…” (line 296)

“…the  presence of sodium alginate carrier exerts a certain hindering effect on the reaction system. the presence of sodium alginate carrier has a certain hindering effect on the ions in the reaction system”. (lines 300-303)

“Notably, the immobilized amylase exhibited…” (line 303)

“Effective reusability has the potential to reduce the demand for free cellulase in industrial production…” (lines 338-340)

“Subsequently” (line 431)

“The active target enzyme can hydrolyz, the two gels were compared, the corresponding target bands were cut, and identity confirmation of the target enzyme by HPLC/MS-MS analysis” (lines 432-434)

Comments on the Quality of English Language

The comments concerning the (low) quality of English are included in my review

Author Response

Thanks for your letter and the reviewers’ comments concerning our manuscript entitled “The isolation, identification and immobilization method of three novel enzymes with diosgenin-producing activity derived from an Aspergillus flavus” with the Manuscript ID: ijms-2714420. We appreciate the valuable suggestions of the reviewers. The comments are very helpful for improving our paper, as well as having important guiding significance to our further work. We have considered the comments carefully and revised the manuscript accordingly. The English syntax and word usage in this manuscript have been improved than before. The whole manuscript has been polished by a native English speaker and improved significantly. The revised manuscript is hereby resubmitted by us now, all changes to the manuscript have been highlighted in red. Main corrections in the paper and the response to the reviewer’s comments are as follows. We hope that all corrections are satisfactory and look forward to your positive decision. Please feel free to contact us with any questions.

Thank you very much for your work concerning our paper.

Wish you all the best!

Sincerely yours

Shirong Feng et al.

Comments 1:  Please delete “technology” from the title. This work reports the results of immobilization experiments, it’s a long way up to a technology.

Response 1: Thank you for pointing this out. We agree with this comment. Therefore, we have changed the title to “The isolation, identification and immobilization method of three novel enzymes with diosgenin-producing activity derived from an Aspergillus flavus.

Comments 2: The introduction part is not appropriate, essential data are missing. The utilization of dioscin glycosidases for the hydrolysis of DZW saponins is mentioned in only one phrase (lines 48-50), indicating 4 references. Please extend this part by adding other references (there are many), like as Xiang et al., 2018 (https://doi.org/10.1007/s00253-018-9030-5), Chen et al., 2018 (https://doi.org/10.1016/j.steroids.2018.05.005), and particularly the recent review of Zhang et al. (https://doi.org/10.1016/j.steroids.2023.109181). Moreover, the results reported by other research groups must be briefly presented, allowing a comparison with their own results (which must be performed in the discussion section).

Response 2: Agree. We have rewritten this part and added relevant references from Line 49 to 71.

Comments 3: In the introduction part, all references (#18 to #21) supporting the limitations of soluble (native) enzymes utilization and their overcoming by immobilization were selected from a single enzyme class, the lipases. Instead, the authors must cite review articles or book chapters that discuss the advantages of immobilizing enzymes (mainly for industrial applications) in a more general manner. The immobilization by entrapment, particularly in alginate matrices, must be also described based on literature data, highlighting its advantages compared to other immobilization methods for the immobilization of the studied enzymes. I recommend completely rewriting and enlarging this part. 

Response 3: Thank you for pointing this out. We agree with this comment. Therefore, I have completely rewritten and enlarging this part. Please refer to the text from Line 74 to 99.

Comments 4: Results and discussion, page 2. Please add a reaction scheme (or a more comprehensive scheme, also including the isolation and immobilization steps) to provide an overview of the accomplished work. Also, please add at the beginning of every subchapter a phrase summarizing and justifying the methodology selected for the experimental work described in that section.

Response 4: Agree. We have added a phrase at the beginning of each line of the result to summarize the work. Please refer to the text from Line 114, Line 130 and Line 138. A flowchart was added to explain in Fig.7.

Comments 5: Page 2, Subchapter “Isolation optimization of dioscin glycosidases” (this title must be changed, maybe to “Isolation of dioscin glycosidases by fractional precipitation with ammonium sulphate”, maybe it would also be useful to number the subchapters as 2.1, 2.2, etc.) Please add a table presenting the activity, specific activity and total activity recovery yield for every isolation and purification step. Otherwise, it is difficult for the reader to evaluate the real efficiency of this approach.

Response 5: Thank you for pointing this out. We agree with this comment. Therefore, we have changed this title to “Isolation of dioscin-glycosidases by fractional precipitation with ammonium sulphate”. Please refer to the text from Line 112. And the tables of the activity, specific activity and total activity recovery yield for every isolation and purification step, as shown in Table S1 to Table S3 in Supplementary material.

Comments 6:  I cannot identify the term “gel imaging autography”. As I understand, N-PAGE was used as a preliminary check of dioscin glycosidase activity in the fraction spots, but please explain it in the text. 

Response 6: Thank you for pointing this out.  We agree with this comment. We have rephrased the term as “enzyme-linked-substrate autography strategy”. Because we are original with this method, we have our own definition of the term. The main process of this method is the N-PAGE in saponin solution (20 mg/mL),then placed on orbital oscillator at 4 °C. The diosgenin will precipitate on the gel to form bands. And the result is the right half of Fig. 2 (a). Please refer to the text from Line 437 to Line 440.

Comments 7: The authors must disclose what exactly means “protein concentration” determined in the presence of so-called “protective agents” and the significance of its determination for the immobilization of the enzyme (being included in the immobilization optimization chapter).

Response 7: Thank you for pointing this out. We agree with this comment. Therefore, the actual meaning of the protein concentration we defined is the enzyme concentration, the specific activity of the enzyme was determined, and has been corrected in the article. Please refer to Line 517-526.

Comments 8: Materials and methods part and throughout the manuscript. Please calculate the enzymatic activity units as micromoles/min, as commonly used, not as micrograms/min. The definition given at page 14, lines 529-531 “1 U is the amount of 1 microgram of diosgenin produced per minute per milliliter of dioscin-glycosidase solution to be tested by reacting with the saponin substrate” is not correct. As it was defined long time ago, the standard unit of enzyme activity is the enzyme amount which will catalyze the transformation of 1 micromole of the substrate per minute under standard conditions (Eur. J. Biochem. 1979, 97, 319-320). So, the activity of the enzyme in liquid form (solution) must be expressed as U/ml, while the activity of a solid enzyme preparate as U/mg.

Response 8: Thank you for pointing this out. We agree with this comment. Therefore, we have corrected the definition of enzyme activity in the paper and re-calculated it. Please refer to Line 495-497. And the results as shown in Supplementary material.

Comments 9: The manuscript contains many syntax, grammar, and spelling errors. Please revise it comprehensively and also be consistent and clear in the discussion part. The following examples represent only a selection, it is impossible for me to discuss all. 

Response 9: Thank you for pointing this out. We agree with this comment. The English syntax and word usage in this manuscript have been improved than before. The whole manuscript has been polished by a native English speaker and improved significantly.

4. Response to Comments on the Quality of English Language

Point 1: The comments concerning the (low) quality of English are included in my review.

Response 1: We have polished by a native English speaker and improved significantly.The English syntax and word usage in this manuscript have been improved than before.

Reviewer 2 Report

Comments and Suggestions for Authors

Review report for ijms-2714420 entitled “The isolation, identification and immobilization technology of three novel enzymes with diosgenin-producing activity derived from an Aspergillus flavus”

The authors focus on enzymatic reactions as a method for synthesizing diosgenin, a raw material for synthesizing steroid drugs, in a clean way, and propose a new method for isolating, identifying, and immobilizing useful enzymes from endophytic bacteria. The content is concise and its conclusions are based on experimental results. It is considered ready for publication after some modifications.

Questions and comments

  1. The reviewer read the authors' previous paper and needed help understanding why the authors focused on SYfx2 13.2. Why did the authors decide to base their research on this endophytic fungus? What advantages does this have over using it in place of other endophytic bacteria? Please add an explanation in the introduction.
  2. The section “Isolation optimization of dioscin-glycosidases” contains the results of diosgenin glycosidase? (P3, line 85). If so, the reviewer recommends to change the title of this section. OR just a typo?
  3. What is the CK in Fig. 3?
  4. The characters in Figs. 4, 5, and 6 better be large.
  5. Enzyme immobilization using sodium alginate and calcium chloride is a classical method, but please tell me why the authors chose this method of immobilization among many methods of enzyme immobilization.
  6. Does the method proposed in this paper have a chance for beating the current mainstream diosgenin production method? I would like to know the specific cost and production efficiency per unit volume, and future prospects.
  7. Typos
    1. P2, Line 74 “increa”

Author Response

Thanks for your letter and the reviewers’ comments concerning our manuscript entitled “The isolation, identification and immobilization method of three novel enzymes with diosgenin-producing activity derived from an Aspergillus flavus” with the Manuscript ID: ijms-2714420. We appreciate the valuable suggestions of the reviewers. The comments are very helpful for improving our paper, as well as having important guiding significance to our further work. We have considered the comments carefully and revised the manuscript accordingly. The English syntax and word usage in this manuscript have been improved than before. The whole manuscript has been polished by a native English speaker and improved significantly. The revised manuscript is hereby resubmitted by us now, all changes to the manuscript have been highlighted in red. Main corrections in the paper and the response to the reviewer’s comments are as follows. We hope that all corrections are satisfactory and look forward to your positive decision. Please feel free to contact us with any questions.

Thank you very much for your work concerning our paper.

Wish you all the best!

Sincerely yours

Shirong Feng et al.

Comments 1: The reviewer read the authors' previous paper and needed help understanding why the authors focused on SYfx2 13.2. Why did the authors decide to base their research on this endophytic fungus? What advantages does this have over using it in place of other endophytic bacteria? Please add an explanation in the introduction.

Response 1: Thank you for pointing this out. We agree with this comment. Therefore, we have discussed the differences between endophytic fungi and endophytic bacteria in the preface of this article. Please refer to the text from Line 65 to 69.

Comments 2: The section “Isolation optimization of dioscin-glycosidases” contains the results of diosgenin glycosidase? (P3, line 85). If so, the reviewer recommends to change the title of this section. OR just a typo?

Response 2: Thank you for pointing this out. We agree with this comment. Therefore, we have changed this title to “Isolation of dioscin-glycosidases by fractional precipitation with ammonium sulphate”. Please refer to the text from Line 112.

Comments 3: What is the CK in Fig. 3?

Response 3: Thank you for pointing this out. we have changed the illustration to “control”. That's a base medium, it contains no metal ions. (Medium formulation: 5.5% sucrose, 0.6% NH4H2PO4, and 26.6% wheat bran)

Comments 4: The characters in Figs. 4, 5, and 6 better be large.

Response 4: Thank you for pointing this out. We agree with this comment. Therefore, we have changed. Please refer to the text in Fig.4,5 and 6.

Comments 5: Enzyme immobilization using sodium alginate and calcium chloride is a classical method, but please tell me why the authors chose this method of immobilization among many methods of enzyme immobilization.

Response 5: Thank you for pointing this out. We agree with this comment. Therefore, we have explained the reason for using this method clearly in Line 82 to 99.

Comments 6: Does the method proposed in this paper have a chance for beating the current mainstream diosgenin production method? I would like to know the specific cost and production efficiency per unit volume, and future prospects.

Response 6: Thank you for pointing this out. We fully believe in our methods have a chance for beating the current mainstream diosgenin production method. For example, in our laboratory conditions. Taking the production of 1000g diosgenin as an example for cost calculation:

a) Acid hydrolysis method: The rhizoma of DZW is first crushed, and then each gram of powder is subjected to hydrolysis at 100°C with the addition of 30mL 3mol/L H2SO4.  Subsequently, a good deal of NaOH is utilized for neutralization, resulting in substantial wastewater that requires treatment. The filtered solids are subsequently dried at 60°C and extracted three times with 20L petroleum ether to obtain 1000g diosgenin.  According to market reagent consumables cost calculation, the total cost amounts to 800 yuan.

b) The methods mentioned herein aim to activate strains, prepare seed solution and fermentation medium. A total of 400g ammonium sulfate is required for DGAs separation. The quantity of sodium alginate and calcium chloride needed for immobilized enzyme preparation is relatively small. Based on market reagent consumables costs, a total expenditure of 540 yuan is necessary to produce 1000g diosgenin. In the actual production process, we can directly react with crude enzyme liquid and saponin to produce saponin, which will save more cost.

At present, the main method of diosgenin production is still acid hydrolysis, but this method uses acid and alkali and generates wastewater, which does not meet the requirements of environmental protection. In this paper, the separation of DGAs from microorganisms and the use of immobilized enzyme technology made diosgenin production more energy efficient, and the cost was reduced by 32.5%.

This work also provides an environmentally friendly, green and energy efficient method to produce diosgenin by mild enzymatic hydrolysis using the three DGAs.  In addition, the enzyme-linked-substrate autography strategy provides a reliable and ef-fective enzyme identification model for the discovery of better DGAs or other specific functional enzymes.

Comments 7: Typos. P2, Line 74 “increa”.

Response 7: Thank you for pointing this out. We agree with this comment. Therefore, we have changed.

Round 2

Reviewer 1 Report

Comments and Suggestions for Authors

Based on the thorough evaluation of the revised manuscript, I consider that the scientific content was improved, indeed. Most recommendations of the reviewers were agreed, and the revisions were made accordingly, with a few exceptions. Because still are some important issues to address, I will reiterate these comments and questions.

1. Response to Comment 8. “We agree with this comment. Therefore, we have corrected the definition of enzyme activity in the paper and re-calculated it. Please refer to Line 495-497. And the results as shown in Supplementary material”. The definition of the enzyme activity was indeed corrected in the manuscript, but the Supplementary material (at least the version I had access) was not modified at all, keeping the same activity values in Tables S1, S2 and S3. It is impossible to obtain the same values when the activity unit (U) is calculated in micromole/min as it was in microgram/min.

2. Response to Comment 9: “The English syntax and word usage in this manuscript have been improved than before. The whole manuscript has been polished by a native English speaker and improved significantly”. It is right that several errors from the original manuscript were corrected. However, mostly in the added part (highlighted in red) there are again many syntax and grammatical errors which make the text confusing and difficult to understand. It is hard to believe  that a native English speaker (with the appropriate technical English expertise) has approved that. Anyway, it must be corrected because in the present form the text is not appropriate, even if the basic content is right.

Comments on the Quality of English Language

My comments regarding the quality of English, which must be consistently improved, are written in my review.

Author Response

Thanks for your letter and the reviewers’ comments concerning our manuscript entitled “The isolation, identification and immobilization method of three novel enzymes with diosgenin-producing activity derived from an Aspergillus flavus” with the Manuscript ID: ijms-2714420. We appreciate the valuable suggestions of the reviewers. The comments are very helpful for improving our paper, as well as having important guiding significance to our further work. We have considered the comments carefully and revised the manuscript accordingly. The English syntax and word usage in this manuscript have been improved than before. The whole manuscript has been polished by a native English speaker and improved significantly. The revised manuscript is hereby resubmitted by us now, all changes to the manuscript have been highlighted in green. Main corrections in the paper and the response to the reviewer’s comments are as follows. We hope that all corrections are satisfactory and look forward to your positive decision. Please feel free to contact us with any questions.

Thank you very much for your work concerning our paper.

Comments 1: Response to Comment 8. “We agree with this comment. Therefore, we have corrected the definition of enzyme activity in the paper and re-calculated it. Please refer to Line 495-497. And the results as shown in Supplementary material”. The definition of the enzyme activity was indeed corrected in the manuscript, but the Supplementary material (at least the version I had access) was not modified at all, keeping the same activity values in Tables S1, S2 and S3. It is impossible to obtain the same values when the activity unit (U) is calculated in micromole/min as it was in microgram/min.

Response 1: Thank you for pointing this out. We agree with this comment. Therefore, we have corrected the contents of the manuscript and Supplementary Materials. Please refer to the text from Line 118 to 162 and Supplementary Materials.

Comments 2:  Response to Comment 9: “The English syntax and word usage in this manuscript have been improved than before. The whole manuscript has been polished by a native English speaker and improved significantly”. It is right that several errors from the original manuscript were corrected. However, mostly in the added part (highlighted in red) there are again many syntax and grammatical errors which make the text confusing and difficult to understand. It is hard to believe that a native English speaker (with the appropriate technical English expertise) has approved that. Anyway, it must be corrected because in the present form the text is not appropriate, even if the basic content is right.

Response 2: Agree. We have accepted the retouching service of MDPI. The following is the proof of embellishment.
